# Ancestry Studies in Forensic Anthropology: Back on the Frontier of Racism

**DOI:** 10.3390/biology10070602

**Published:** 2021-06-29

**Authors:** Ann H. Ross, Shanna E. Williams

**Affiliations:** 1Human Identification & Forensic Analysis Laboratory, Department of Biological Sciences, North Carolina State University, Raleigh, NC 276995, USA; 2Department of Biomedical Sciences, UofSC School of Medicine Greenville, Greenville, SC 29605, USA; WILL3992@greenvillemed.sc.edu

**Keywords:** race, ancestry, population affinity, craniofacial variation, geometric morphometrics

## Abstract

**Simple Summary:**

Within the practice of forensic anthropology ancestry is oftentimes used as a proxy for social race. This concept and its implications were explored via a content analysis (2009–2019) of the Journal of Forensic Sciences. Our findings revealed antiquated views of race based on the trifecta of continental populations (Asia, Europe, and Africa) continue to be pervasive in the field despite scientific invalidation of the concept of race decades earlier. Moreover, our employment of modern geometric morphometric and spatial analysis methods on craniofacial coordinate anatomical landmarks from several Latin American samples produced results in which the groups were not patterned by ancestry trifecta. Based on our findings we propose replacing the assumption of continental ancestry with a population structure approach that combines microevolutionary and cultural factors with historical events in the examination of population affinity.

**Abstract:**

One of the parameters forensic anthropologists have traditionally estimated is ancestry, which is used in the United States as a proxy for social race. Its use is controversial because the biological race concept was debunked by scientists decades ago. However, many forensic anthropologists contend, in part, that because social race categories used by law enforcement can be predicted by cranial variation, ancestry remains a necessary parameter for estimation. Here, we use content analysis of the *Journal of Forensic Sciences* for the period 2009–2019 to demonstrate the use of various nomenclature and resultant confusion in ancestry estimation studies, and as a mechanism to discuss how forensic anthropologists have eschewed a human variation approach to studying human morphological differences in favor of a simplistic and debunked typological one. Further, we employ modern geometric morphometric and spatial analysis methods on craniofacial coordinate anatomical landmarks from several Latin American samples to test the validity of applying the antiquated tri-continental approach to ancestry (i.e., African, Asian, European). Our results indicate groups are not patterned by the ancestry trifecta. These findings illustrate the benefit and necessity of embracing studies that employ population structure models to better understand human variation and the historical factors that have influenced it.

## 1. Introduction

Forensic anthropology is a sub-discipline of biological anthropology, the science of studying what it means to be human via our biology. Forensic anthropologists are experts in human skeletal anatomy, growth, and development; expertise that we use in medicolegal death investigations for the recovery and analysis of human skeletal remains. A significant part of our analysis is the creation of the biological profile, an evaluation of four criteria that may assist with identification: age-at-death, sex (for adult skeletons), stature, and ancestry [1]. The estimation of ancestry is one of the most difficult (and controversial) parameters of the biological profile. It is often conflated with social race and ethnicity by medical examiners, law enforcement, forensic practitioners, and government agencies. Further, some practitioners have questioned the validity of estimating this parameter and if the estimation could even hinder identification because of racial bias on the part of investigative agencies [2,3,4]. Part of the reason its use is so controversial is that the biological race concept, namely, that the human species can be divided into biological races, was debunked decades ago [5]. In the 1990s there was discord within biological anthropology stimulated by a paper by Lieberman and colleagues [6], presented earlier in 1987 at the American Association of Physical Anthropologists annual meeting that reported 50% of the biological anthropologists polled believed in the race concept. Forensic anthropologists argued that it was a pragmatic decision to include “race” in their forensic case reports as “race” was used by law enforcement and medicolegal death investigators working the missing and unidentified cases [7]. Thus, in 1992 a name change from “race” to “ancestry” was proposed as a less loaded term [7]. This has been rationalized by the notion that we can connect craniofacial morphology (i.e., size and shape variants of skull bone features) to social race categories (e.g., United States Census categories) [8,9]. However, some biological anthropologists questioned the ethics of even estimating this parameter fearing that its continued use would endorse racist views and be complicit in the social injustices faced by underrepresented groups [2,10,11,12]. 

In a search for the term “ancestry” in the titles of the *Journal Forensic Sciences (JFS)* between the years of 2009-2019, 20 articles used “ancestry” and in 2010 and 2011, two articles still used “race.” The term ancestry appeared 24 times in the keywords between 2009-2019, with four papers using samples identified as *black*, *white*, and *Hispanic*. Five papers used samples identified as *black* and *white*, which included a paper on *South African blacks* and *whites*. There were 12 papers with various iterations of “*Hispanic*” (i.e., *South West Hispanics*); as well as papers that defined their samples as *Prehistoric Native Americans*; those that use a few country names (e.g., Japanese, Guatemala, Germany, Thailand, etc.); and a paper on *Native American*, *Japanese*, and *Thai* samples. This literature review clearly illustrates the lack of purpose, consensus, and consistent usage of the nomenclature; suggesting that the transition from race to ancestry was primarily a linguistic change (see [13] that covers the problems with nomenclature). The many iterations of “Hispanic” are a result of the 2008 migrant death symposium at the American Academy of Forensic Sciences annual meeting dealing with the difficulty of identifying unidentified border-crossers (UBCs) in the United States. Interestingly, the term Hispanic is still commonly used even though it has no biological meaning [14], and going as far back as 1992, pioneering forensic anthropologist Alice Brues understood that “Hispanics” from South Florida, the Southwest, and Texas should not be grouped under one umbrella because they represented different population migrations to the US [14,15].

The results of this literature review also illustrate the return to antiquated and over-simplistic views of race based on the trifecta of continental populations from Asia, Europe, and Africa used by typologists of the early 20th century, have regained popularity [16]. In part, this is because the reference databases we rely on to compare cranial measurements of an unknown person were constructed using such categories. However, this facile presumption ignores underlying microevolutionary mechanisms such as drift, migrations, and mutation that are responsible for human variation and diversity. Studies of global populations reveal that human craniofacial morphology fits a neutral evolutionary model because contiguous populations more frequently exchange genes and/or share common ancestry [17].

Therefore, rather than studying population affinity via an assumption of continental ancestry, we instead advocate for a population structure approach. The benefit of such an approach is that it allows us to understand how microevolutionary factors such as genetic drift act in concert with cultural factors (i.e., marriage patterns) and historical events (i.e., epidemics, colonization) to influence human variation. A population structure approach is empirically driven, meaning that it is based on firm observations without phylogenetic assumptions and by operational approaches that are hypothesis-driven by meaningful questions [18]. When comparing populations one can select various types of characters for investigation such as morphology, physiology, behavior, and/or ecology. However, common mistakes made in the selection of a character for estimating similarity is a failure to identify the biological factors that the characters represent (i.e., their heredity) and assumptions that they are all equally informative in providing evidence of group (i.e., phenetic) similarity [18]. One example of the former is with the use of the skull trait variant post-bregmatic depression [3,4]. As noted, a major consideration in the application of a population structure approach is to account for historical events such as population influxes and settlements, religious secularization, language differences, temporality, and spatial patterning that would be impacted by microevolutionary forces [19].

Myopically, forensic anthropology abandoned the study of human biological variation based on a strong foundation of examining human variation through a population structure lens grounded in microevolution, and instead re-embraced a typological approach that looks a lot like “race” of the early years [20,21]. Therefore, it is clear that a broad synthesis to better understand the underlying patterns of modern human variation that would disclose the underlying population structure of the group(s) under study is needed. Such information would also be of use to biological anthropology more broadly. Here, we use craniofacial coordinate anatomical landmarks from Latin American samples while employing modern geometric morphometric and spatial analysis methods to test the validity of applying the antiquated tri-continental approach to ancestry. These samples were chosen given the stated problems with the comprehensive, non-critical use of the “*Hispanic*” label for anyone from Latin America or Spain, and in an attempt to partition out how different historical socio-political events within Latin America have influenced biological variation. Further, we discuss how situating such approaches within a microevolutionary framework can enrich our understanding of how major historical events influence human variation and population structure.

## 2. Materials and Methods

### 2.1. Samples

The sample totals 397 modern adult individuals and includes individuals from Latin America (Chile, Colombia, Cuba, Guatemala, Panama, Puerto Rico, and Peru); and comparative skeletal samples from Spain and enslaved Africans from Cuba were included to explore the effects of colonialism and the Transatlantic Slave Trade on the population structure of the region. Males and females were analyzed separately when this information was available (see Table 1). Some samples were small due to poor preservation in tropical environments. To incorporate all of the observed biological information and to increase sample sizes males and females were pooled as it has been found that sex variation is negligible within each population included in population [22]. Latitude and longitude were recorded based on present-day political boundaries. The sample composition is presented in Table 1.

While we acknowledge the value data collected from such samples continue to contribute to discussions of human variation, it should also be noted that the history and ethics of human skeletal collections, in general, is often dubious. Such body harvesting all too often occurred under the umbrella of scientific racism, without the permission of the deceased or next of kin, and disproportionately targeted marginalized populations.

Sixteen type 1 and 2 standard anatomical craniofacial landmarks (for a total number of landmarks 16 × 3 dimensions = 48) that should reflect the among-group variation were utilized in the analyses (Table 2 and Figure 1). The landmarks selected were those that are of particular interest in forensic anthropology and that would allow for broader shape coverage. To mitigate the effect of small sample sizes, a PCA was used as a dimension-reducing technique and limiting the number of variables [23,24].

### 2.2. Landmark Precision and Reliability

Only type 1 and type 2 landmarks were included as they have been found to be reliably reproducible [25]. The landmarks included are those that were found to meet the less than 5 percent error threshold for digitizing and intra-observer error [25]. The coordinate data were collected using a Microscribe G2X digitizer with a reported average error rate of 239 mm [26]. These samples are part of the reference database for the classification software 3D-ID [27] and prior to inclusion in the software, data underwent extensive error checks via mapping (i.e., visualization) of all individuals using the Generalized Procrustes analysis or GPA function in Morpheus et al. [28].

### 2.3. Geometric Morphometrics

Before statistical analyses can be performed, coordinate data must first undergo a GPA transformation using the software *MorphoJ*, which is freely available for downloading and developed by Klingenberg [29]. GPA brings all specimens into a common coordinate system, after it translates, rotates, and scales each individual. The advantage of this method is that morphological shape and size can be examined separately, with shape defined as all of the geometric information that remains after the effects of location, scale, and rotational effects are removed [30,31]. Centroid size is defined as a measure of geometric scale that is mathematically independent of shape [31]. To reduce the dimensionality, a principal component analysis (PCA) of the covariance matrix was performed on the GPA-transformed coordinate data and these principal components were utilized for ensuing statistical analyses [31]. A canonical variates analysis (CVA) was performed to examine the most amount of the variation with the least dimensions possible of the a priori groups [29]. A generalized distance measure (or Mahalanobis distance) was used to examine group similarity [29]. A discriminant function analysis (DFA) was performed to visualize morphological variation between the consensus configurations of each group. The phenetic (e.g., morphological) among-group variation was examined using ANOVA for centroid size. Among-group variation for shape was analyzed using MANOVA of the principal components scores derived from MorphoJ. The ANOVA and MANOVA procedures were performed in JMP^®^ Pro 14.1 [32].

### 2.4. Hierarchical Clustering

Average linkage hierarchical (or agglomerative) clustering was conducted using the generalized distance matrix to examine group similarity [33,34]. The process begins with each population sample in a single cluster, then in each successive iteration, it merges the closest pair of clusters until all the data is in one cluster. The cluster analysis was performed in JMP^®^ Pro 14.1 [32].

### 2.5. Spatial Analysis

Moran’s *I*, a product-moment coefficient, was used to measure the spatial autocorrelation of shape (PC1 as only one variable can be utilized) and centroid size, which is a measure of genetic similarity between individuals with reference to geographic separation (latitude/longitude). Spatial correlograms were computed to evaluate the spatial autocorrelation coefficients for all pairs of localities at specified geographic distance classes [35], and were performed using the freeware software GeoDa v1.14.0 [36].

## 3. Results

### 3.1. Geometric Morphometrics

Forty-one PC scores were generated from the covariance matrix, which were used as new variables in the subsequent statistical analyses. The ANOVA shows that size is significantly different among the groups (Centroid size: (F (11, 385) = 22.35, *p* ≤ 0.0001). The MANOVA (of 41 principal component scores derived by MorphoJ) also detected significant shape variation (Shape: Wilks’ Lambda 0.0058, df = 451, 3706.6, F = 5.12, *p* ≤ 0.0001). The anatomical landmarks used here are in the same location on each skull; this property enables evaluation and observation of any distinctions in overall cranial shape and size between groups. Morphological variation is illustrated via wireframe graphs that depict the magnitude and direction of shape change between two mean configurations with the direction of change depicted from light (light blue) to dark (blue). The starting shape is that of one sample mean configuration that is deformed into a target shape (second sample) mean configuration to visualize the differences. The groups illustrated were selected according to the clusters produced by the hierarchical cluster analysis. The similarity between the Chilean male mean configuration (light blue) and the Spanish male mean configuration (blue) is visualized showing little to no variation in the placement of the anatomical landmarks (Figure 2).

To illustrate the importance of a population approach, Panama and Colombia, Panama and enslaved Africans, and Panama and Spanish consensus configurations were compared based on known historical events (i.e., conquest, colonialism, and slavery). The morphological differences between the Colombians and the Panamanians show that the Colombians (light blue) have shorter and narrower crania than Panamanians (blue), depicted by the more posteriorly and inferiorly placed anatomical landmarks bregma and lambda and more superiorly placed anatomical landmarks asterion and zygomaxillare (Figure 3). It also shows that Colombians have a longer upper facial height with the anatomical landmark nasion positioned more superiorly and a more inferiorly placed anatomical landmark zygomaxillare. Enslaved Africans (light blue) have longer and narrower cranial vaults with anatomical landmark lambda more posteriorly placed and asterion more anteriorly placed compared to Panamanians (blue). The wireframe depicting the starting shape of Panamanians (light blue) shows that they have shorter cranial vaults and a shorter and more projecting upper face as evidenced by the more anteriorly placed anatomical landmarks subspinale, bregma, and lambda, and more inferiorly positioned anatomical landmarks bregma and zygomaxillare than the Spaniards’ target shape (blue), see Figure 3.

### 3.2. Hierarchical Clustering

The dendrogram produced from the hierarchical cluster analysis using the generalized distance matrix shows two distinct clusters: (1) Chile/Spain and (2) Panama, Cuba, Guatemala, and Colombia which branch off the Chile/Spain cluster. The enslaved African sample clusters with Peru, and Puerto Rico is the most dissimilar. This is further illustrated by the constellation plot (Figure 4), which arranges the samples as endpoints. The length of a line between cluster joints represents the distance between them. The plot shows that the most distinct group is the sample from Puerto Rico, which is three times the distance from the Colombian samples and closest to Peru and enslaved African samples. Chileans and Spaniards are closer to each other than to the rest of the groups.

### 3.3. Spatial Analysis

The spatial autocorrelation for shape (using PC1 accounting for 21 percent of the total variance) and size show that the groups are spatially patterned and heterogeneous indicated by the positive and significant Z-scores (Table 3). While the correlograms show the autocorrelations decreasing with increased distance, the pattern is generally non-monotonic, meaning the pattern is not clinal as would be expected under an isolation-by-distance model such as kinship [35], for both shape and centroid size. Autocorrelations are expected to be positive at closer distances and negative at greater distances (Figure 5). The correlograms do not support a clinal pattern.

## 4. Discussion

Even though forensic anthropology as a discipline has moved away from using the term “race” to that of “ancestry”, the early critics of race estimation in forensics questioned whether the underlying approach to ancestry would change. Thirty years have passed since this initial criticism and as evidenced by the research published during this time period, ancestry studies have not advanced past the typological (see for example [37]). It is also clear that current research is not fundamentally grounded in an evolutionary framework to understand what has shaped modern human craniofacial [3,4]. The studies surveyed as part of our content analysis show an over-simplistic, typological, tri-continental approach that underscores the need for a paradigm shift to a population structure approach, which incorporates the study of population affinity to understanding modern human biological variation. This paradigm shift can be applied through meaningful hypotheses and avoiding thoughtless comparisons of one sample to another without purpose (e.g., Thai to European Americans, etc.) and by utilizing non-racialized and appropriate reference samples in forensic classification software. For example, implementing nomenclature changes and sample selection in existing commonly used forensic software such as Fordisc [38], which uses inconsistent terms such as “White, Black, Hispanic, Guatemala, and Japanese”, which reflect continental-level, biologically meaningless, and/or country labels; and AncesTrees [39], which uses prehistoric samples that are not applicable for forensic use with antiquated six race categories based in typology, would be a good path forward.

In a recent regional population structure study of pre-contact New World craniofacial variation, Ross and Ubelaker [40] demonstrated that craniofacial variation was a complex interplay between the environment and microevolutionary forces and not the result of a single mechanism. They demonstrated that generally, these pre-Contact populations were spatially patterned, consistent with an isolation-by-distance model. However, they also found a weak association between shape-related variation and altitude, and climate. In the present study, a similar population structure approach was applied to modern Latin American samples to test whether the antiquated trifecta approach to ancestry was valid. Our results demonstrate that Puerto Rico is the most different from the others; Spain and Chile are the most similar to each other compared to the other samples; Panama, Cuba, Guatemala, and Colombia link to the Spain and Chile cluster; and Peru and enslaved Africans form a separate cluster.

The Spanish conquistadors brought enslaved Africans with them beginning as early as 1501 to the Caribbean coast of Panama to colonize the New World [41]. Before the arrival of the Spaniards, there were an estimated 25,000 Amerindians in Panama; by 1522 their population estimates were 13,000 [41]. As a result of the decimation of these Indigenous populations resulting from epidemics and warfare, the Spaniards forced migrations of neighboring Indigenous populations from Panama and Nicaragua; and during Pizarro’s expedition to Peru in 1527, 10,000 Amerindians were forcibly displaced to Peru [41]. The association of the Spanish and Chilean samples can be therefore explained through the complex history of conquest and colonialism.

The city of Santiago, Chile was founded in 1542 by Spanish conquistador Pedro de Valdivia. However, the Spanish conquest of Chile was delayed by a long war with Auracanian Indians [42]. During the colonial period, entire Indian populations were decimated by disease and forced labor [42]. From the time of European arrival, slavery of abducted Africans was present, primarily on the Caribbean coast of South America (e.g., Venezuela and Colombia) and in Ecuador and Peru, as well as [42]. Recent work focused on La Isabela, the settlement established after Christopher Columbus’ second voyage to what is now the Dominican Republic, suggests that at least one person of African origin was present [43]. The influence of the Transatlantic Slave Trade was detected here by the hierarchical cluster analysis linking Peru and the enslaved African samples. The constellation plot further elucidates the relationship among the groups and illustrates that while the sample from Puerto Rico is the most dissimilar, it is closest to the Peru-enslaved African cluster, followed by Colombia, Guatemala, Cuba, and Panama—all depicting early contact with the Spanish conquistadors that brought enslaved Africans. The spatial analysis was used to assess if there was a spatial pattern based on geographic location. While Moran’s *I* was significant and positive for both shape and size, the correlograms show that they are not clinal. The morphological variation for pre-contact populations suggests heterogeneity from the initial population diffusion into the New World prior to European contact [40]. While there is a morphological spatial pattern of modern Latin Americans they do not show a monotonic decrease with distance, but rather indicate repeated population migrations and expansions such as European colonization, the Transatlantic Slave Trade, and forced migrations of Indigenous groups [44]. The argument that there are no races, only clines (or a neutral evolutionary model because neighboring populations more frequently exchange genes and/or share a common ancestry) is not supported here. This finding illustrates a more complex mechanism of modern craniofacial variation and underscores the need for applying a population structure and evolutionary lens to the practice of forensic anthropology. 

We use Panama with its complicated history, which has been coveted since the Spanish conquest for its geographic feature as a land bridge of the American continents between the Atlantic and Pacific Oceans, to illustrate the complex nature of assessing population affinity in forensic practice. During the Spanish colonial period, jurisdiction for the Panama territory passed from the Viceroyalty of Lima to Bogotá in the 18th century; it finally gained independence from Spain in 1821 but was part of the Republic of Colombia until 1903 [41]. Importantly, before Panama’s split from Colombia, in 1847, a United States merchant set out to build a railroad across the Isthmus that would combine land and sea and open up the Pacific [45,46]. During its construction, a workforce was brought from across the globe (e.g., Austria, China, Colombia, England, France, Germany, India, Ireland, and Jamaica) with thousands dying of malaria, yellow fever, and hardships from the tropical environment [47]. Another important milestone after the failed attempt by the French in the late 1800s was the enormous federally funded undertaking by the United States from 1904–1914 to build an interoceanic canal, a massive earthwork project the likes of which had never been attempted [40,47].

These trans-isthmus ventures brought thousands of migrant workers (~60% from the West Indies) to Panama. The racial contrast of the workers to the engineers and project leaders is crucial to understanding the societal organization and marginalization in the Panama Canal Zone [40]. The colonial caste system transformed into the rigid racial categories imposed by the United States in the Panama Canal Zone, which segregated the workforce both physically and geographically. The Panama Canal Zone was a socialist experiment divided by the white elite minority and the West Indian majority. European Americans showed open disdain for the Panamanians which combined with a culture of flagrant inequality inherited from Spain [40,47]. This segregation, an apartheid not witnessed in any other 20th-century Latin American country [40], was still unmistakable as late as 1986 when the first author graduated from secondary school in the former Zone. Given the complexity of Panama’s history, our results are therefore not surprising when viewed against this backdrop. An analysis that rather solely focused on rigid ancestral categories would not have been able to pinpoint Panamanians’ dissimilarity to neighboring countries, in particular to Colombia with their shared history under colonial rule. In modern forensic anthropology, all of these heterogeneous groups would have been erroneously designated under the label “Hispanic.”

The results of the present study demonstrate that there is substantial diversity in Latin American populations, typically organized into the biologically meaningless grouping of “Hispanic” in contemporary forensic practice. Furthermore, this study obviates the rejection of the tricontinental approach to ancestry estimation and underscores the need for applying a population structure approach with an evolutionary lens to not only understand factors that have influenced craniofacial morphology but test hypotheses about population movements and the impact of major historical events such as conquest and slavery.

## 5. Conclusions

In 2000, Smay and Armelagos [2] stated that “it was interesting that the word race was being replaced by the less provocative term ancestry”, while also indicating they doubted that the logic behind race would change and that the analysis of races using exclusive categories based on folk taxonomy would continue simply under a different name—they were right. Ancestry has become a synonym for race. Given our current global political climate, continuing to type individuals in this way lends credence to existing power structures and socioeconomic inequalities. A mere word change is like putting lipstick on a pig, an ineffective attempt at beautifying and obfuscating something whose unsightly features are still evident. We need a fundamental, structural, and thoughtful shift in our paradigm beginning with hypotheses driven by meaningful questions and careful selection of informative characters for investigation. We need a return—or rather, beginning—to investigating real human biological variation.

## Figures and Tables

**Figure 1 biology-10-00602-f001:**
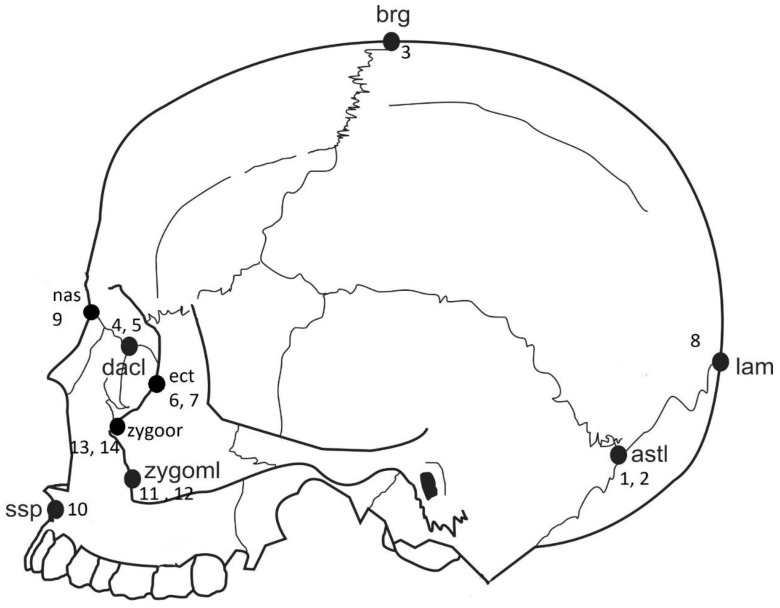
Anatomical landmark location and associated landmark number from Table 1.

**Figure 2 biology-10-00602-f002:**
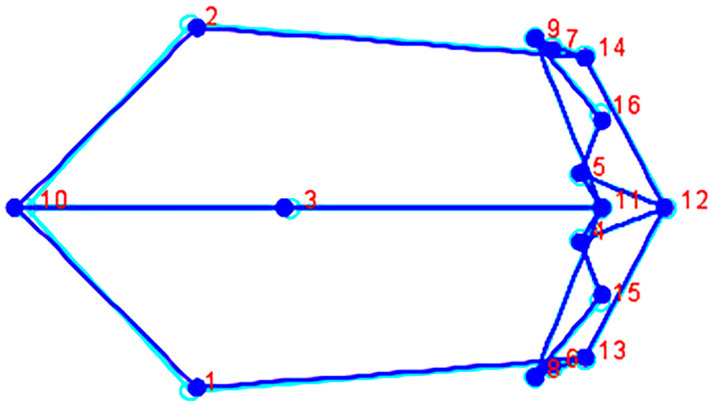
Wireframe (superior view) depicting the Chilean male mean configuration (starting shape, light blue) deformed into the Spanish male mean configuration (target shape, blue). The numbers correspond to the landmarks in Table 2.

**Figure 3 biology-10-00602-f003:**
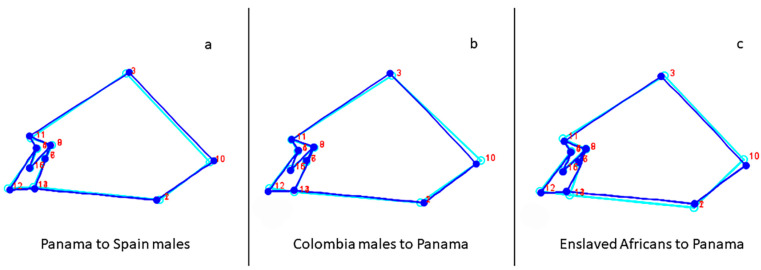
Wireframes depicting the (**a**) Panama (light blue) into Spanish males (blue); (**b**) Colombian males (light blue) into Panama (blue); (**c**) Enslaved Africans (light blue) into Panama (blue). Numbers correspond to landmarks in Table 2.

**Figure 4 biology-10-00602-f004:**
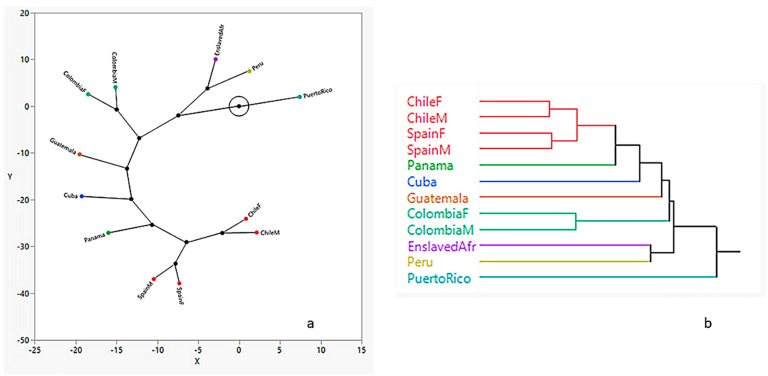
Constellation plot (**a**) and dendrogram (**b**) results from hierarchical cluster analysis showing group relationships.

**Figure 5 biology-10-00602-f005:**
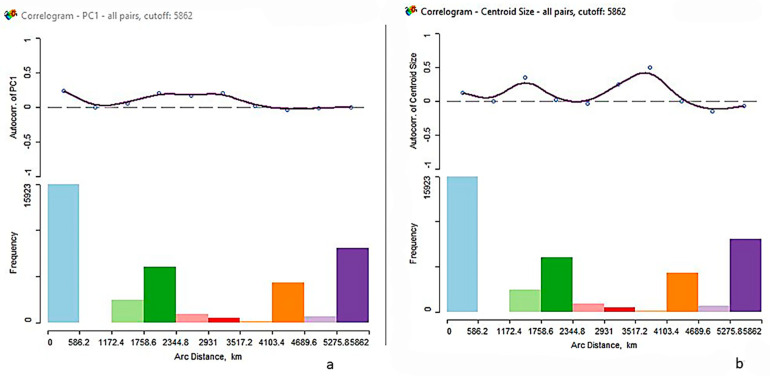
Correlograms for shape (**a**) and size (**b**) depicting the spatial autocorrelation. Moran’s *I* by distance in kilometers.

**Table 1 biology-10-00602-t001:** Sample composition and provenience.

Group	N	Provenance	Latitude	Longitude
Chile	♀ = 34 ♂ = 37	Juan Munizaga Collection, Universidad de Chile, Santiago, Chile	−33.45	−70.67
Colombia	♀ = 11 ♂ = 53	Antioquia modern skeletal collection, Escuela Nacional de Criminalística, Medellín, Colombia	6.230833	−75.5906
Cuba	19	Cemetery Collection, Museo de Montane, Havana, Cuba	23.11359	−82.3666
Enslaved Africans	25	Morton Collection, University of Pennsylvania, US	−8.83833	13.23444
Guatemala	♂ = 71	Provided by Kate Spradley	14.62843	−90.5227
Puerto Rico	♂ = 5	University of Rio Piedras, Puerto Rico	18.46633	−66.1057
Panama	10	Insituto de Medicina Legal, Panama	8.983333	−79.5167
Peru	7	C.A. Pound Human ID Lab, University of Florida, US	−12.0464	−77.0428
Spain	♀ = 58 ♂ = 67	Oloriz Collection, Madrid, Spain	40.41678	−3.70379

**Table 2 biology-10-00602-t002:** Anatomical landmarks and associated numbers.

Landmark Number	Anatomical Landmarks
1, 2	Asterion, bilateral
3	Bregma
4, 5	Dacryon, bilateral
6, 7	Ectoconchion, bilateral
8,9	Frontomalare temporale, bilateral
10	Lambda
11	Nasion
12	Subspinale
13, 14	Zygomaxillare, bilateral
15, 16	Zygoorbitale, bilateral

**Table 3 biology-10-00602-t003:** Moran’s *I* results for shape using the first principal component and size using centroid size with reference to geographic location.

Moran’s *I*	Observed	Expected	Std Dev	Z	PR > Z
PC1	0.0695	−0.0025	0.0022	32.5	0.001
CS	0.0027	−0.0025	0.0026	2.0	0.027

## Data Availability

Data a contained within the article.

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
