# Peer review of "Ancestry Studies in Forensic Anthropology: Back on the Frontier of Racism"

_biology, 2021, doi:10.3390/biology10070602_

Round 1

Reviewer 1 Report

Thank you for the opportunity to review this paper on the problems associated with ancestry studies in forensic anthropology.  The authors address this seeming name change from race to ancestry in the 1990s and highlight that there was not also a change in approach to research, which was needed.  They provide data to show that human variation is not clinal and that careful consideration is needed when formulating hypotheses and in defining groups of study.  This move to a population approach will encompass evolutionary theory and move away from typology.  

I enjoyed reading the paper and think that the subject matter is timely and well argued.  I only have very minor suggestions to clarify some analyses and to contextualize the findings just a bit more.

Specific Edits/comments:

Intro:
I appreciated the discussion of the problems associated with collections acquisition.  I feel like this is something that we need to be much more open about in our papers.  And, will likely be a more common discussion in the future.

Methods:
How was sexual dimorphism in cranial metrics handled? - Males and females are generally kept separate - but, in a few collections males and females aren't separated (e.g., enslaved Africans) - are those both males and females?  - this just needs a bit of explanation since there is sexual dimorphism in cranial metrics and the data should be treated appropriately to handle it (and, I hope I didn't just miss it in there already!)

Discussion:
Could you explore the non-clinal finding a bit more and why that is significant? Specifically because it goes against the idea that "there are no races, only clines" - it is much more complex than that as you have illustrated with population history and evolutionary approach - variation should be discontinuous.  You do make this point, but I think you could add a couple more sentences since it seems critical to your argument.

The population approach advocated here also moves away from social race terms at least as defined by the US census, which may be worth mentioning - so it doesn't seem like you are arguing just for a name change,  it is a recognition of a new approach.  Or at least addressing more thoroughly how the approach advocated here is not the same as equating skeletal metrics to social race.

minor edits:
I think the preference is to capitalize Indigenous

Author Response

Methods:

How was sexual dimorphism in cranial metrics handled? - Males and females are generally kept separate - but, in a few collections males and females aren't separated (e.g., enslaved Africans) - are those both males and females?  - this just needs a bit of explanation since there is sexual dimorphism in cranial metrics and the data should be treated appropriately to handle it (and, I hope I didn't just miss it in there already!)

  • Thank you for pointing this out. Sexes were pooled to maximize biological variation and to increase sample sizes for the smaller samples. In inter-population comparisons, it has been found that sex variation is negligible within each population. We have included a reference.

Discussion:

Could you explore the non-clinal finding a bit more and why that is significant? Specifically because it goes against the idea that "there are no races, only clines" - it is much more complex than that as you have illustrated with population history and evolutionary approach - variation should be discontinuous.  You do make this point, but I think you could add a couple more sentences since it seems critical to your argument.

  • A few sentences have been added to the Discussion to address this issue. We agree that it strengthens the paper. Thank you for the feedback.

The population approach advocated here also moves away from social race terms at least as defined by the US census, which may be worth mentioning - so it doesn't seem like you are arguing just for a name change,  it is a recognition of a new approach.  Or at least addressing more thoroughly how the approach advocated here is not the same as equating skeletal metrics to social race.

  • Reference to the US Census has been added to the Introduction

minor edits:

I think the preference is to capitalize Indigenous

  • All references to Indigenous have been capitalized.

Reviewer 2 Report

See pdf file

Author Response

Very few things to say about this part, for which I am not specialist. The first paragraph sets the framework. The authors cite previous studies at the end of the paragraph, but not in the beginning (lines 36 to 44), when they mention some theoretical basements, and some definitions. I think that some references would help.

  • : A foundational forensic anthropology textbook citation (Langley and Tersigni-Tarrant, 2017) has been added to the introduction to contextualize the theoretical framework this paper is based upon.

At the end of this part (lines 102 to 110), the authors repeat ideas already mentioned (line 53). The same thing for the idea lines 118, very similar to the one lines 45.

  • Thank you for pointing out these unintentional redundancies. We have streamlined them within the Introduction.

Then, the link between the main concepts and ideas and the application to the

study of Latin American is not obvious. The authors could rework this part to make it more fluid and logical.

  • The authors have reorganized the document, in particular the introduction, to clarify the connection between the paper’s concepts and the utilization of Latin American studies.

Material and Methods:

I see many methodological problems in this part. If the workflow is globally correct, some points need precisions.

-First, some comments about the sample composition. This sample is not balanced. The number of males doesn't balance the number of females and there is huge differences between populations. How would it affect the results ? I know it is an easy comment and I know samples are never easy to constitute, but I would really like the authors to have a clear consideration of this possible effect.

  • We appreciate the insight. Unfortunately, sample sizes from tropical environments are small due to poor preservation. To mitigate this issue, we use PCA as a dimension-reducing technique and use the PC scores as new variables in subsequent statistical analyses. We have included language in the methods and results section.

-  The number of landmarks is very low, only 14. As a morphometrician, I know that modern morphometrics can provide new tools to increase the number of landmarks. I am not forcing the authors to add landmarks and semilandmarks just on principle. I know how useful can be a simple study when it works. However, in the case of shape analysis of human skulls, a great part of the shape (braincase) is not taken into account. Most of the landmarks are located on the face. These problems were standing in the early XXIth and authors such as Philipp Gunz developed solutions to capture the shape without landmarks.

  • We appreciate these comments. Because we use coordinate data the actual number of variables is calculated as 14x3(number of dimensions) or 42 landmarks. We have clarified this in the manuscript.
  • We only included type 1 and 2 landmarks as type 3 landmarks (encompassing most of the vault) have a lot of error associated with them. While the vault is of interest in studies of our fossil ancestors, in modern humans, and due to the plasticity of the vault we included landmarks that are best suited for population structure studies. We also added a reference.

-  the authors never mentioned an absolutely necessary point. The measurement error. To me, it is not possible to publish any geometric morphometrics study without estimation of error. I really advise the authors to work on this point. I would recommend the Bailey and Byrnes (1990) estimation.

  • Thank you for bringing this up. We use standard craniofacial landmarks and only type 1 and type 2 that are highly repeatable. In addition, as all this data is included in the software 3D-ID it underwent extensive data checks and validation prior to inclusion. In GM unlike traditional morphometrics, it is not common to provide an estimation of error because of the nature of the data. We included language and a citation of accuracy and error for the Microscribe of a reported average error of .239mm.

-  it is not very clear to me how many PCs the authors analyse after GPA and PCA. It seems that they mainly analyse the first PC.

  • Thank you for pointing this out and we have clarified that there were 22 PC scores extracted accounting for 100% variation, which were utilized in the subsequent analyses.

I would recommend to analyse the entire variation, that is to say the 3k-7 non null PCs.

  • See above response

It is not clear neither what the authors consider as groups in the Geometric morphometrics part. Is it population, is it sex????

  • The groups are those listed in table 1 and depicted in the hierarchical cluster analysis.

Results:

The results part is clear but, here again, I have some questions.

First, what are the values of df linel92? I understand the values of df for the first test (11 and 385 but I am not sure to understand the values for the second test (what is 242, and what is 15.785???).

  • Thank you for bringing this to our attention- we have corrected it in the manuscript on how to arrive at the F value. For centroid size the 11 is the individual df and the second number is the residuals df, for shape 242 is the individual df, and 15785 is the residuals df. (Centroid size: (F(11, 385) = 22.35 , p = <.0001); Shape: (F(242, 15,785) = 6.82, p = <.0001)

Looking at shape variation on figure 3 makes me react another time about shape coverage. The authors indicate variation between populations, mainly expressed in the posterior part of skull. It would describe differences in the braincase shape, which is the less described part.

I am very surprised by the results of the clustering. Looking at the variation shown on figure 3, I can't understand the perfect ordering of the populations in the tree, especially for populations with males and females. The effect of sexual dimorphism is very low in comparison with the population effect. This is surprising. I would suspect a more mixed signal.

  • In inter-population comparisons, it has been found that sex variation is negligible within each population- we have clarified this and have added a citation.
  • I would not consider the agglomerative cluster to be perfectly ordered. Figure 3 depicts shape change, while the clustering is based on the Mahalanobis/generalized distance matrix. The process begins with each population sample in a single cluster, then in each successive iteration, it merges the closest pair of clusters until all the data is in one cluster.

Discussion:

The discussion, is nice, and well written. The ideas are clear. The first paragraph develops idea of high importance .

The other paragraphs set the context of the population structure and are very informative. The authors, and their results, really add something for the understanding of the population structure.

  • Thank you for your comments.

However, I have one question. I understand that, looking at the clustering, the authors demonstrate  that the Latin American  populations show substantial  diversity  (which would contribute to reject the Hispanic label), but this same clustering show a perfect ordering of the populations that could be seen as real delimitations. How do the authors understand what could be seen as a paradox????

I understand  that that spatial  correlation  doesn't  work, I  understand  that the tricontinental approach doesn't fit the model, but this perfect ordering would imply true differences  between  populations  rather than  mixes. What is the  opinion  of the  authors on this point?

  • We do not see clustering as perfect ordering. All new world populations are related to some extent from the original population expansions- the variation observed relates to later historical events.

Round 2

Reviewer 2 Report

Ancestry Studies in forensic anthropology: back on the frontier of racism

Ann H. Ross, Shanna E. Williams

Biology-1245625

This is the second review of the manuscript.

I still have some minor comments but he manuscript could be published after these small corrections. Thus, I would like to thank the authors for the work they have done to correct and improve the manuscript.

Material and Methods:

I see many methodological problems in this part. If the workflow is globally correct, some points need precisions.

-First, some comments about the sample composition. This sample is not balanced. The number of males doesn't balance the number of females and there is huge differences between populations. How would it affect the results ? I know it is an easy comment and I know samples are never easy to constitute, but I would really like the authors to have a clear consideration of this possible effect.

  • We appreciate the insight. Unfortunately, sample sizes from tropical environments are small due to poor preservation. To mitigate this issue, we use PCA as a dimension-reducing technique and use the PC scores as new variables in subsequent statistical analyses. We have included language in the methods and results section.

- I appreciate these precisions, but this is an important issue. What about BGPCA methods for instance? It is supposed to reduct this effect. It could be, at least mentioned.

- The number of landmarks is very low, only 14. As a morphometrician, I know that modern morphometrics can provide new tools to increase the number of landmarks. I am not forcing the authors to add landmarks and semilandmarks just on principle. I know how useful can be a simple study when it works. However, in the case of shape analysis of human skulls, a great part of the shape (braincase) is not taken into account. Most of the landmarks are located on the face. These problems were standing in the early XXIth and authors such as Philipp Gunz developed solutions to capture the shape without landmarks.

  • We appreciate these comments. Because we use coordinate data the actual number of variables is calculated as 14x3(number of dimensions) or 42 landmarks. We have clarified this in the manuscript.
  • I really don ‘t agree with the authors on this point. I understand that 14x3=42, I understand they have 42 variables, but it is still 14 landmarks on a complex structure and I still think it is not enough to cover the entire shape variation. If the authors say it is enough to get the main signal, OK, but to me the coverage is low.
  • We only included type 1 and 2 landmarks as type 3 landmarks (encompassing most of the vault) have a lot of error associated with them. While the vault is of interest in studies of our fossil ancestors, in modern humans, and due to the plasticity of the vault we included landmarks that are best suited for population structure studies. We also added a reference.

- same comment. I don’t agree. To me if the neurocranium is out of interest, then why do the authors used landmarks on it? Moreover, they are probably wrong because shape variation can be seen on the neurocranium (vault and occipital part).

- the authors never mentioned an absolutely necessary point. The measurement error. To me, it is not possible to publish any geometric morphometrics study without estimation of error. I really advise the authors to work on this point. I would recommend the Bailey and Byrnes (1990) estimation.

  • Thank you for bringing this up. We use standard craniofacial landmarks and only type 1 and type 2 that are highly repeatable. In addition, as all this data is included in the software 3D-ID it underwent extensive data checks and validation prior to inclusion. In GM unlike traditional morphometrics, it is not common to provide an estimation of error because of the nature of the data. We included language and a citation of accuracy and error for the Microscribe of a reported average error of .239mm.

- GM studies, just like all quantitative studies MUST have an estimation of measurement error. This is not the average error reported by the device, this is the error made when acquiring the dataset. To me this is absolutely something to add if the authors want to be trusted.

- it is not very clear to me how many PCs the authors analyse after GPA and PCA. It seems that they mainly analyse the first PC.

  • Thank you for pointing this out and we have clarified that there were 22 PC scores extracted accounting for 100% variation, which were utilized in the subsequent analyses.
  • NO, I cannot believe this point. In 3D geometric Morphometrics, 100% variation is reached at 3k-7 PCS. Here they have 14 landmarks, the authors have 100% of variation at 42-7=35 PCS, not more, not less. Thus I don’t understand the 22 Pcs estimation.

Results:

The results part is clear but, here again, I have some questions.

First, what are the values of df linel92? I understand the values of df for the first test (11 and 385 but I am not sure to understand the values for the second test (what is 242, and what is 15.785???).

  • Thank you for bringing this to our attention- we have corrected it in the manuscript on how to arrive at the F value. For centroid size the 11 is the individual df and the second number is the residuals df, for shape 242 is the individual df, and 15785 is the residuals df. (Centroid size: (F(11, 385) = 22.35 , p = <.0001); Shape: (F(242, 15,785) = 6.82, p = <.0001)
  • Sorry but I still have a problem with the df. Still ok with 11 and 385 but where does the 242 comes from??? Df are based on numbers of variables. 11 is 12 (nb of groups)-1, 385 is 397 (number of specimens)-12, what is 242? Probably 11 (nb of groups minus 1) *22 (nb of PCS)???? I really don’t understand 115,785??? Could you please give the readers some indications?

Author Response

-First, some comments about the sample composition. This sample is not balanced. The number of males doesn't balance the number of females and there is huge differences between populations. How would it affect the results ? I know it is an easy comment and I know samples are never easy to constitute, but I would really like the authors to have a clear consideration of this possible effect.

  • We appreciate the insight. Unfortunately, sample sizes from tropical environments are small due to poor preservation. To mitigate this issue, we use PCA as a dimension-reducing technique and use the PC scores as new variables in subsequent statistical analyses. We have included language in the methods and results section.

- I appreciate these precisions, but this is an important issue. What about BGPCA methods for instance? It is supposed to reduct this effect. It could be, at least mentioned.

RESPONSE:

Yes, PCA was used to mitigate the issue of small sample sizes and limiting the number of variables.

- The number of landmarks is very low, only 14. As a morphometrician, I know that modern morphometrics can provide new tools to increase the number of landmarks. I am not forcing the authors to add landmarks and semilandmarks just on principle. I know how useful can be a simple study when it works. However, in the case of shape analysis of human skulls, a great part of the shape (braincase) is not taken into account. Most of the landmarks are located on the face. These problems were standing in the early XXIth and authors such as Philipp Gunz developed solutions to capture the shape without landmarks.

  • We appreciate these comments. Because we use coordinate data the actual number of variables is calculated as 14x3(number of dimensions) or 42 landmarks. We have clarified this in the manuscript.
  • I really don ‘t agree with the authors on this point. I understand that 14x3=42, I understand they have 42 variables, but it is still 14 landmarks on a complex structure and I still think it is not enough to cover the entire shape variation. If the authors say it is enough to get the main signal, OK, but to me the coverage is low.

RESPONSE

We incorrectly reported 14 landmarks when we used 16.  This was a typo left over from a conference presentation. 

Sixteen type 1 and 2 standard anatomical craniofacial landmarks (for a total number of landmarks 16x3 dimensions = 48) that should reflect the among-group variation were utilized in the analyses (Table 2 and Figure 1).  The landmarks selected were those that are of particular interest in forensic anthropology and that would allow for broader shape coverage.  To mitigate the effect of small sample sizes, a PCA was used as a dimension reducing technique and limiting the number of variables (Cardini et al., 2019, Rohlf, 2021).

  • We only included type 1 and 2 landmarks as type 3 landmarks (encompassing most of the vault) have a lot of error associated with them. While the vault is of interest in studies of our fossil ancestors, in modern humans, and due to the plasticity of the vault we included landmarks that are best suited for population structure studies. We also added a reference.

same comment. I don’t agree. To me if the neurocranium is out of interest, then why do the authors used landmarks on it? Moreover, they are probably wrong because shape variation can be seen on the neurocranium (vault and occipital part).

RESPONSE

Sixteen type 1 and 2 standard anatomical craniofacial landmarks (for a total number of landmarks 16x3 dimensions = 48) that should reflect the among-group variation were utilized in the analyses (Table 2 and Figure 1).  The landmarks selected were those that are of particular interest in forensic anthropology and that would allow for broader shape coverage.  To mitigate the effect of small sample sizes, a PCA was used as a dimension reducing technique and limiting the number of variables (Cardini et al., 2019, Rohlf, 2021).

- the authors never mentioned an absolutely necessary point. The measurement error. To me, it is not possible to publish any geometric morphometrics study without estimation of error. I really advise the authors to work on this point. I would recommend the Bailey and Byrnes (1990) estimation.

  • Thank you for bringing this up. We use standard craniofacial landmarks and only type 1 and type 2 that are highly repeatable. In addition, as all this data is included in the software 3D-ID it underwent extensive data checks and validation prior to inclusion. In GM unlike traditional morphometrics, it is not common to provide an estimation of error because of the nature of the data. We included language and a citation of accuracy and error for the Microscribe of a reported average error of .239mm.

GM studies, just like all quantitative studies MUST have an estimation of measurement error. This is not the average error reported by the device, this is the error made when acquiring the dataset. To me this is absolutely something to add if the authors want to be trusted.

RESPONSE

We appreciate the reviewer’s comments.  Ross and Williams (2008) tested repeatability and error in coordinate landmark data. Our landmarks are those that meet the less than 5% error threshold for digitizing and intra-observer error.

- it is not very clear to me how many PCs the authors analyse after GPA and PCA. It seems that they mainly analyse the first PC.

  • Thank you for pointing this out and we have clarified that there were 22 PC scores extracted accounting for 100% variation, which were utilized in the subsequent analyses.
  • NO, I cannot believe this point. In 3D geometric Morphometrics, 100% variation is reached at 3k-7 PCS. Here they have 14 landmarks, the authors have 100% of variation at 42-7=35 PCS, not more, not less. Thus I don’t understand the 22 Pcs estimation.

RESPONSE

We thank the reviewer for bringing this up. In light of this very important critique. We re-ran the analyses in MorphoJ without the “symmetry” option, which was done inadvertently.  41 PCs were derived from the new run.

Results:

The results part is clear but, here again, I have some questions.

First, what are the values of df linel92? I understand the values of df for the first test (11 and 385 but I am not sure to understand the values for the second test (what is 242, and what is 15.785???).

  • Thank you for bringing this to our attention- we have corrected it in the manuscript on how to arrive at the F value. For centroid size the 11 is the individual df and the second number is the residuals df, for shape 242 is the individual df, and 15785 is the residuals df. (Centroid size: (F(11, 385) = 22.35 , p = <.0001); Shape: (F(242, 15,785) = 6.82, p = <.0001)
  • Sorry but I still have a problem with the df. Still ok with 11 and 385 but where does the 242 comes from??? Df are based on numbers of variables. 11 is 12 (nb of groups)-1, 385 is 397 (number of specimens)-12, what is 242? Probably 11 (nb of groups minus 1) *22 (nb of PCS)???? I really don’t understand 115,785??? Could you please give the readers some indications?

RESPONSE

We profusely thank the reviewer for catching this.  After re-running the analyses, we opted to export the PCs and re-ran an ANOVA for centroid size and a MANOVA for shape (41 PCs) in SAS JMP.  We are grateful to the reviewer for their thorough review as this otherwise may have gone unnoticed. 

Forty-one PC scores were generated from the covariance matrix, which were used as new variables in the subsequent statistical analyses. The ANOVA shows that size is significantly different among the groups (Centroid size: (F(11, 385) = 22.35 , p = <.0001). The MANOVA (of 41 principal component scores derived by MorphoJ) also detected significant shape variation (Shape: Wilks’ Lamda 0.0058, df = 451, 3706.6, F = 5.12, p = <.0001).